# Biocontrol Potential of an Endophytic *Pseudomonas poae* Strain against the Grapevine Trunk Disease Pathogen *Neofusicoccum luteum* and Its Mechanism of Action

**DOI:** 10.3390/plants12112132

**Published:** 2023-05-28

**Authors:** Jennifer Millera Niem, Regina Billones-Baaijens, Benjamin J. Stodart, Pierluigi Reveglia, Sandra Savocchia

**Affiliations:** 1Gulbali Institute, Charles Sturt University, Locked Bag 588, Wagga Wagga, NSW 2678, Australia; rbaaijens@csu.edu.au (R.B.-B.); bstodart@csu.edu.au (B.J.S.); preveglia@ias.csic.es (P.R.); ssavocchia@csu.edu.au (S.S.); 2Faculty of Science and Health, School of Agricultural, Environmental and Veterinary Sciences, Charles Sturt University, Locked Bag 588, Wagga Wagga, NSW 2678, Australia; 3UPLB Museum of Natural History, University of the Philippines Los Baños, College, Los Baños 4031, Laguna, Philippines; 4Institute of Weed Science, Entomology, and Plant Pathology, College of Agriculture and Food Science, University of the Philippines Los Baños, College, Los Baños 4031, Laguna, Philippines; 5Institute for Sustainable Agriculture, CSIC, 14004 Córdoba, Spain

**Keywords:** *Vitis vinifera* cv. Shiraz, cyclic lipopeptide, rifampicin resistance, antagonistic bacteria, qPCR

## Abstract

Grapevine trunk diseases (GTDs) impact the sustainability of vineyards worldwide and management options are currently limited. Biological control agents (BCAs) may offer a viable alternative for disease control. With an aim to develop an effective biocontrol strategy against the GTD pathogen *Neofusicoccum luteum*, this study investigated the following: (1) the efficacy of the strains in suppressing the BD pathogen *N. luteum* in detached canes and potted vines; (2) the ability of a strain of *Pseudomonas poae* (BCA17) to colonize and persist within grapevine tissues; and (3) the mode of action of BCA17 to antagonize *N. luteum*. Co-inoculations of the antagonistic bacterial strains with *N. luteum* revealed that one strain of *P. poae* (BCA17) suppressed infection by 100% and 80% in detached canes and potted vines, respectively. Stem inoculations of a laboratory-generated rifampicin-resistant strain of BCA17 in potted vines (cv. Shiraz) indicated the bacterial strain could colonize and persist in the grapevine tissues, potentially providing some protection against GTDs for up to 6 months. The bioactive diffusible compounds secreted by BCA17 significantly reduced the spore germination and fungal biomass of *N. luteum* and the other representative GTD pathogens. Complementary analysis via MALDI-TOF revealed the presence of an unknown cyclic lipopeptide in the bioactive diffusible compounds, which was absent in a non-antagonistic strain of *P. poae* (JMN13), suggesting this novel lipopeptide may be responsible for the biocontrol activity of the BCA17. Our study provided evidence that *P. poae* BCA17 is a potential BCA to combat *N. luteum*, with a potential novel mode of action.

## 1. Introduction

The biocontrol of plant pathogens has been intensified these recent years, particularly the use of beneficial microorganisms, among which are those described as endophytes [1,2,3,4]. The environmental pollution caused by the excessive use and misuse of agrochemicals and the hazard that these chemicals pose to human health and the ecosystem has led to strict regulations on synthetic pesticide use and the banning of the most hazardous from the market. This has paved the way for the discovery of alternative methods for controlling plant diseases. Among those alternatives is biological control. Biological control is the “decrease of inoculum or the disease-producing activity of a pathogen accomplished through one or more organisms” [5].

Due to the limited number of management options available for the control of grapevine trunk diseases (GTDs), biocontrol agents (BCAs) have been explored as a viable alternative. Studies have been conducted on the use of bacteria for the control of grapevine trunk diseases (GTDs) [6,7,8,9,10,11,12]. *Bacillus subtilis*, for instance, has shown antagonistic activity against GTD pathogens *Eutypa lata* [6,7,8] and *Neofusicoccum parvum* [9,10]. A study conducted by Trotel-Aziz et al. in 2019 [9] reported that *B. subtilis* strain PTA-271 protected grapevine cuttings against *N. parvum* by inducing the immune responses of the host plant and by detoxifying the fungal toxins terremutin and R-mellein. *Erwinia herbicola* [8] demonstrated biocontrol activity against *E. lata* in the laboratory although it has subsequently had little success in the field. Actinobacteria were also shown to be promising in controlling GTD pathogens in grafted nursery plants [11]. An endophytic strain, *Streptomyces* sp. VV/E1, and two rhizosphere *Streptomyces* isolates, *Streptomyces* sp. sp. VV/R1 and *Streptomyces* sp. VV/R4, also significantly reduced young grapevine decline infection by the fungal pathogens *Dactylonectria* sp., *Ilyonectria* sp., *Phaeomoniella chlamydospora*, and *Phaeoacremonium minimum* [9]. Another strain, *Streptomyces* sp. E1 + R4, suppressed black-foot disease of grapevine caused by *Dactylonectria torresensis*, *D. macrodidyma*, *Ilyonectria liriodendri*, and *D. alcacerensis* [12]. The use of *Pseudomonas* spp. as BCAs has been widely studied within various pathosystems, including potato scab [13], late blight [14], Rhizoctonia root rot on bean [15], damping-off and root rot in tomato [16], anthracnose in beans [17], post-harvest diseases of stone fruit [18], and take-all disease of wheat [19,20,21]. In grapevines, *Pseudomonas* spp. were shown to suppress *Botrytis cinerea* [22,23], root-rot pathogens [24], and *Rhizobium vitis*, which causes crown gall [25]. *Pseudomonas* spp. Were found to be present in the grapevine endosphere [26,27,28]. The presence of *Pseudomonas* spp. in the inner grapevine tissues may be an indication of their ability to colonize and survive *in planta*.

Recently, the antagonistic activity of 10 endophytic strains of *Pseudomonas* from grapevines was demonstrated against the pathogens associated with Botryosphaeria, Eutypa dieback, and Esca/Petri disease [29]. The in vitro inhibition of these pathogens may imply that these antagonistic strains are potential BCAs. However, the demonstrated inhibition of a pathogen in vitro does not always translate to effective biocontrol in vivo. A successful BCA must also be able to survive in the host plant and efficiently colonize the plant tissues for a sufficient period to provide protection [13,30]. Hence, *in planta* assays to investigate the efficacy of antagonistic strains of *Pseudomonas* against GTD pathogens in grapevines and their colonization potential of the plant is essential.

Plant protection against pathogen infection using *Pseudomonas* spp. is mediated by a variety of secondary metabolites, among which are antibiotics such as 2,4-diacetylphloroglucinol (DAPG) [21,31,32,33] and phenazine and its derivatives [13,34,35,36]. These antibiotics play a role in the induction of systemic resistance, biofilm formation, and membrane permeability. Beneficial fluorescent *Pseudomonas* spp. also enhance plant growth and exert antagonistic activity against plant pathogens by producing siderophores [37,38,39], which cause a reduction in iron availability for other microorganisms, including plant pathogens, resulting in less efficient iron-sequestering systems [40,41]. Cyclic lipopeptides (CLPs) are also produced by many strains of *Pseudomonas* spp. [42,43,44,45]. CLPs can insert into membranes and perturb their function, which results in broad antibacterial and antifungal activities [46]. Some CLPs also act as a biosurfactant, providing the organism improved surface colonization and utilization of surface-bound substrates [47]. The production of volatile organic compounds [33,48,49,50], niche exclusion [51,52], production of lytic enzymes, [53] and induction of systemic resistance [54,55,56,57,58,59,60] have also been implicated in the plant-protecting capacity for different strains of *Pseudomonas*. Thus, the mechanism of action involved in the control of GTD pathogens by these *Pseudomonas* strains requires further investigation. Understanding their mode of action may help determine their capacity to efficiently suppress disease in natural environments.

This study aims to determine the potential of an antagonistic *Pseudomonas poae* strain (BCA17) to control the GTD pathogen *N. luteum*. The objectives are as follows: (1) to investigate the ability of *Pseudomonas* spp. to suppress *N. luteum* in detached canes and potted grapevines; (2) to demonstrate the endophytic colonization of grapevine tissues by a rifampicin-resistant BCA17 strain in pre-planted cuttings and potted vines; and (3) to investigate a potential direct mechanism of BCA17 in the control of GTD pathogens and disease.

## 2. Results

### 2.1. Suppression of Neofusicoccum luteum Infection in Planta by Pseudomonas Strains

Of the 10 antagonistic strains of *Pseudomonas* inoculated on detached canes, six significantly reduced the recovery of *Neofusicoccum luteum* (*p* ≤ 0.005), while only BCA17 resulted in no pathogen recovery 4 weeks post-inoculation of conidia (Figure 1). Pathogen recovery in canes treated with BCA12, BCA15, and BCA20 (56–78%) was not significantly different from that in the positive control canes (100%) inoculated with *N. luteum* only.

At 3 months post-inoculation in potted grapevines, BCA17 resulted in the lowest percentage recovery of *N. luteum* at 20% (*p* ≤ 0.001); BCA13 and BCA14 resulted in 70% and 90% recovery; and *N. luteum* was not recovered from the negative control vines, which were uninoculated but treated with sterile distilled water (SDW) (Figure 2).

#### Assessment of *N. luteum* Infections by qPCR

A quantitative PCR assay of tissue samples from potted vines treated with strains of *Pseudomonas* prior to inoculation with *N. luteum* confirmed the previous results. The amount of Botryosphaeriaceae DNA detected in vines treated with BCA17 was significantly lower, with 300 copies of DNA detected compared to 12,000 for infected vines not treated with BCA (*N. luteum* only), a 40-fold reduction (Figure 3). Grapevine plants treated with BCA13 and BCA14 did not significantly reduce *N. luteum* with an average of 3000–14,000 copies of Botryosphaeriaceae DNA detected. Pathogen DNA was not detected in any of the uninoculated vines, indicating that there was no background infection in the plant material due to the Botryosphaeriaceae species.

### 2.2. Pseudomonas Colonization and Establishment in Grapevine Tissues

#### 2.2.1. Pre-Planting Application of *Pseudomonas* RifMut Strain to Canes

The analysis of canes treated for 24 h with the rifampicin-resistant (RifMut) strain of *Pseudomonas poae* BCA17 prior to rooting demonstrated that *Pseudomonas* spp. can be reisolated from the five randomly selected canes, revealing that the bacterial strain was translocated into the grapevine canes (Figure 4). The RifMut strain was reisolated in all five wood pieces obtained from the base and middle sections (approximately 13–15 cm distance) of the canes, while only two wood pieces of the top end of the canes were positive to RifMut (Figure 4). Bacteria were not re-isolated from canes dipped in Ringer’s solution only.

Subsequent re-isolations post-rooting showed the RifMut strain still persisted in the grapevine tissues at 8 weeks post-treatment. The RifMut strain was present in all the 10 rooted vines from the base and middle sections, while the RifMut strain was only re-isolated from three vines of the top sections at this time-point of analysis (Figure 4). Bacteria were not recovered from the control canes in Ringer’s solution at post-rooting.

#### 2.2.2. Post-Planting Application of *Pseudomonas*

The assessment of potted grapevine plants inoculated with RifMut shows that at one month after inoculation RifMut was reisolated up to 2.4 and 2.8 cm below and above the inoculation point, respectively, from all sub-sampled vines (Figure 5). Such acropetal and basipetal movements of the RifMut strain were also observed at 3 months after inoculations. At 6 months, the entire plant was colonized by RifMut for the five sampled vines. The bacterial strain was reisolated from the apical tip of the living tissue down to the base of the trunk below the ground. Subsequent re-isolation on lateral shoots also indicates lateral movement of RifMut up to a distance of 3 cm in the shoot sections. Bacteria were not recovered from control plants inoculated with SDW only.

### 2.3. Bioactivity of P. poae BCA17 Culture Filtrate on Mycelial Growth and Spore Germination of GTD Pathogens

#### 2.3.1. Effect of BCA17 Culture Filtrate on Mycelial Growth

The cell-free culture filtrates derived from BCA17 were inhibitory to the mycelial growth of *Diplodia seriata* DAR79990, *N. luteum* DAR81016, *N. parvum* DAR80004, and *Eutypa lata* WB052, regardless of whether these were obtained from cultures co-inoculated with the pathogen or not (Figure 6a–d). The mycelial weight of *D. seriata* DAR79990 grown in NB only was significantly higher (76.26 mg; *p* ≤ 0.01) when compared to mycelia grown in the culture filtrate of BCA17, where mycelial biomasses of 5.23 mg (filtrate obtained from BCA17 + pathogen co-culture) and 1.87 mg (filtrate obtained from BCA17 only culture) were obtained (Figure 6a). A similar pattern was observed with *N. luteum* DAR81016 with 181.7 mg of mycelial biomass (Figure 6b) and *N. parvum* DAR80004 with 190.63 mg in filtrates (Figure 6c) containing NB only. *N. luteum* DAR81016 and *N. parvum* DAR80004 did not grow in the culture filtrate of BCA17 (Figure 6b,c). *E. lata* WB052 was typically slow growing and the mycelial weight in NB was 15 mg, which was significantly different from the biomass of the pathogen that was grown in the culture filtrate of BCA17 (*p* ≤ 0.01) where no growth of the pathogen was observed after 7 days (Figure 6d). No significant differences in fungal biomass (*p* ≥ 0.05) were observed between culture filtrates derived from broth cultures with or without the pathogen (Figure 6a–d).

#### 2.3.2. Effect of BCA17 Culture Filtrate on Spore Germination

The spore germination of *D. seriata* DAR79990, *N. luteum* DAR81016, *N. parvum* DAR80004, and *E. lata* WB052 was inhibited by the culture filtrates of BCA17 at varying degrees 24 h after incubation. The BCA17 culture filtrate reduced *D. seriata* DAR79990 spore germination to 1%, which was significantly lower (*p* ≤ 0.0001) than the untreated control at 60% germination (Figure 7a). The BCA17 culture filtrate also reduced *N. luteum* DAR81016 (Figure 7b) and *N. parvum* DAR80004 (Figure 7c) spore germination with 27 and 13% germination, respectively, which was significantly lower than the untreated control at 48 and 47% spore germination, respectively. *E. lata* WB052 was the least sensitive to the culture filtrate resulting in 86% germination in the control treatment and 77% in the BCA17 culture filtrate (Figure 7d).

### 2.4. Identification of Secondary Metabolite Biosynthetic Gene Clusters

The secondary metabolites and their corresponding biosynthetic gene clusters are summarized in Appendix A. The genome features of the antagonistic *P. poae* strains BCA13, BCA14, and BCA17 and the non-antagonistic *P. poae* strain JMN1 were very similar. Genome analysis using antiSMASH 4.0 [61] identified the consistent presence of gene clusters coding for the synthesis of the siderophore pyoverdine and APEs (aryl polyenes) Vf and Ec, which comprised the largest biosynthetic gene cluster (BGC). The four BCA strains of *P. poae* also displayed a notable array of BGC associated with the production of various cyclic lipopeptides (CLPs), including rhizoxin, viscosin, orfamide, putisolvin, poaeamide, bananamide, entolysin, anikasin, white-line-inducing principle (WLIP), cichopeptin, tolaasin, sessilin, xantholysin, and arthrofactin (Appendix A). Some of these CLPs have previously been described to have an antifungal property.

### 2.5. Identification of the Bioactive Lipopeptides Produced by BCA17

Due to the abundance of CLP-associated genes within the strains of *P. poae*, a complementary analysis was undertaken to verify which lipopeptides are specifically produced by BCA17, and therefore could be involved in the biocontrol activity of the antagonistic *P. poae* BCA17. Utilizing MALDI-TOF MS, a possible lipopeptide with a [M + H]^+^ mass to charge ratio at 2098.2913 *m*/*z* was detected in *P. poae* BCA17 (Figure 8) but was absent in *P. poae* JMN1 (Figure 9). A comprehensive assessment of literature articles indicated that the [M+H]^+^ ion at 2098.2913 *m*/*z* is likely a new lipopeptide; however, the sequence of the compound could not be confirmed with MS/MS analysis due to a lack of specific MS data within online databases.

From de novo sequencing, the partial protein sequence was confirmed to be leu-val-gln-leu-val-val-gln-leu-val (LVQLVVQLV) (C48H87N11O12) (Figure 10), with a monoisotopic mass of 1009.654. This finding does not directly match with the literature [62,63,64,65,66], but a ring opening reaction with ammonia confirmed that the unknown lipopeptide is cyclic in nature. Therefore, and in accordance with the literature, this new lipopeptide is likely related to or belongs to the group of tolaasin. The partial identified sequence shares 8 out of 9 amino acids with the L4VSLVVQLV12 sequence of tolaasin with only 1difference (a Glu instead of the Ser present in tolaasin) [66,67].

## 3. Materials and Methods

### 3.1. Plant Materials

*Vitis vinifera* (cv. Shiraz) was used for all *in planta* experiments. One-year-old dormant canes with no apparent GTD symptoms were collected from grapevines in the Hilltops and Riverina regions, New South Wales (NSW), Australia, in the winters of 2017 and 2019, respectively. All canes were stored at 4 °C for 4–6 weeks until rooting or used in the detached cane assays. For rooting, canes were callused in moist perlite in heat mats maintained at ~30 °C under ambient, winter conditions until roots developed. Rootlings with well-developed roots were transferred to 10 L pots filled with commercial garden soil (compost, 60%; washed sand, 20%; screened loam, 20%). All vines were maintained in the glasshouse (17–27 °C).

### 3.2. Bacterial Strains, Culture Conditions, Inoculum Preparation

The 10 strains of *Pseudomonas* (nine from the *Pseudomonas poae* group and one closely related to *P. moraviensis*) that inhibited the mycelial growth of GTD pathogens in dual culture tests [29] were selected for *in planta* assays. Bacterial cells were obtained from single colonies of each antagonistic strain and stored in 20% glycerol at −80 °C.

To prepare the bacterial inoculum, a loopful of the bacterial suspension in glycerol was streaked onto nutrient agar (NA, Oxoid Ltd., Hampshire, England) and incubated at 25 °C for 2 days. A loopful of the subsequent bacterial growth was then transferred to nutrient broth (NB, Oxoid Ltd.) and incubated for 24 h in an OM15 orbital shaking incubator (RATEK Instruments Pty. Ltd., Victoria, Australia) at 25 °C, 150 rpm. The optical density of the bacterial broth culture was measured at 600 nm (UNICAM 8625 UV/VIS spectrometer, UNICAM Limited, Cambridge, UK) and the bacterial density was adjusted to 10^10^ CFU, which was further confirmed by dilution plating.

### 3.3. Fungal Isolates

*Neofusicoccum luteum* was used as a model pathogen for all the challenge inoculation experiments. Grapevine isolates of *N. luteum* (DAR 80983, 81013, 81016) were used for inoculating both detached canes and potted vines. All isolates were previously identified by sequence analyses of the ITS, EF1-α, and β-tubulin genes [61]. The three isolates were cultured on potato dextrose agar (PDA, Difco Laboratories, Detroit, MI, USA) amended with chloramphenicol (PDAC, 100 mg/L of agar) and incubated at 25 °C overnight in the dark before being transferred under UV light (Phillips 20 w UVB) and exposed to a 12 h light–dark regime at 18–22 °C for 4–5 weeks until fruiting bodies developed. To harvest the conidia, fruiting bodies were scraped and ground with a sterile mortar and pestle before sieving using two layers of sterile Miracloth (Merk Millipore). The conidial suspensions from each of the three isolates were combined and standardized to 1 × 10^4^ conidia/mL and 20 µL (200 conidia) was used for the subsequent inoculations. The viability of the spores was determined by serial dilution plating at three replicate plates.

In studying the antagonistic mechanisms of *P. poae* in vitro, four GTD fungal pathogens were used: *Diplodia seriata* DAR79990, *N. parvum* DAR80004, *N. luteum* DAR81016, and *Eutypa lata* WB052 were used. The fungal pathogens were originally isolated from grapevines exhibiting symptoms of GTD in Australian vineyards and were previously identified by a partial sequencing of the ribosomal RNA gene [68,69,70].

### 3.4. Challenge Inoculation

#### 3.4.1. Detached Canes

The 10 antagonistic strains of *Pseudomonas* (29) were initially screened for their ability to suppress GTD infection using detached canes. Dormant one-year-old Shiraz canes were surface-sterilized with a 0.5% sodium hypochlorite solution for 1 min and rinsed twice with tap water before being cut into single nodes (~10 cm). The fresh wound at the apical end of each cane was inoculated with 20 µL of bacterial suspension (10^10^ CFU) for each of the 10 antagonistic strains [29]. One hour after inoculation, the wounds were subsequently inoculated with the conidia (200 conidia) of *N. luteum*. Canes inoculated with conidia only served as positive controls, while the negative control canes were inoculated with sterile distilled water (SDW). Canes were placed in 5 mL plastic containers containing tap water and incubated for 4 weeks at 25 °C under a 12:12 dark–light regime. For each treatment, three replicate canes were used.

After 4 weeks, all canes were collected, peeled and, surface sterilized in 2.5% sodium hypochlorite and rinsed twice with SDW. The top 2 cm of the cane where the inoculated point was located was excised, cut in half, and transferred onto PDAC. The plates were incubated at 25 °C in darkness and visually assessed for a yellow pigment in the media typical of *N. luteum* for 7–14 days. Recovery of *N. luteum* was evaluated in the following manner: 0-*N, luteum* was not recovered in both cane sections; 1-*N, luteum* was recovered in only 1 section; 2-*N, luteum* was recovered in both cane sections.

#### 3.4.2. Potted Grapevines

The three strains identified as *P. poae* (BCA13, BCA14, and BCA17), which resulted in the lowest pathogen recovery in the detached cane assay, were further used for challenge inoculation using glasshouse potted vines. Apparently healthy 1-yr-old rootlings (cv. Shiraz, ~40 cm) were grown in 10 L pots filled with commercial garden soil (compost, 60%; washed sand, 20%; screened loam, 20%) and maintained in the glasshouse (17–27 °C). The plants were grown and allowed to establish in the potted soil for 3 months prior to inoculation.

In the summer of 2018, inoculations were performed on fresh wounds that were created by cutting the top node of each vine and 20 µL of the BCA suspension (10^10^ CFU) was pipetted onto the wounds as described in detached cane assays. After 1 h, mixed conidial suspensions of three *N. luteum* isolates (DAR 80983, 81013, 81016) were prepared as described previously and 20 µL of the suspension (200 conidia) was applied onto the same inoculation point. Positive control vines were inoculated with conidia of *N. luteum* only, while the negative control vines were inoculated with SDW. Ten replicate vines per treatment were used.

At 3 months after inoculation, the top end for each vine (~2 cm) where the inoculation point was located was collected. Each section was peeled, and surface sterilized as previously described, then cut longitudinally into quarters. Two of the quarters were transferred onto PDAC and incubated at 25 °C in darkness and assessed for growth of N. luteum for 5–7 days as described in Section 3.4.1. The remaining two quarters were placed in sterile 2 mL tubes and stored at −80 °C. The frozen wood samples were subsequently lyophilized and used for DNA extractions and quantitative PCR (qPCR) analysis.

The efficacy of BCA13, BCA14, and BCA17 to suppress *N. luteum* colonization in potted vines was further assessed by qPCR. DNA was extracted from lyophilized wood samples (100 mg) following the methods of Pouzoulet et al. [71] as modified by Niem et al. [29]. All DNA samples were quantified using a Quantus™ Fluorometer (Promega, Madison, WI, USA) prior to qPCR. The qPCR assay was performed in a RotorGene 6000 system (Corbett Life Science, Qiagen, Germantown, MD, USA) using the Botryosphaeriaceae multi-species primers Bot-BtF1 (5′-GTATGGCAATCTTCTGAACG-3′) and Bot-BtR1 (5′-CAGTTGTTACCGGCRCCAGA-3′) and the hydrolysis probe Taq-Bot probe (5′-/56- FAM/TCGAGCCCG/ZEN/GCACCATGGAT/3IBkFQ/-3′) [70]. The PCR amplification was carried out in a 20 µL reaction containing 10 µL of 2× GoTaq^®^ Master Mix (Promega, Madison, WI, USA), 500 nm each of the primers Bot-BtF1 and Bot-BtR1, 250 nm of Taq-Bot probe, 5 µL of template DNA, and nuclease-free water (Qiagen, Germantown, MD, USA) with four technical replicates per sample. Standard Bot-Btub gBlock (500 pg) and a non-template control were included in the assay as positive and negative controls, respectively. The qPCR analysis was performed following the conditions optimized by Billones-Baaijens et al. [70] with the following thermal cycling conditions: 2 min at 98 °C, followed by 40 cycles of 98 °C for 10 s, 60 °C for 30 s, and fluorescence detection at 60 °C for 30 s.

The amount of pathogen DNA amplified was interpolated from a previously developed standard curve [70]. The imported standard curve was calibrated using the standard Bot-Btub gBlock (500 pg) (Integrated DNA Technologies, Inc., Coralville, IA, USA). The number of Botryosphaeriaceae β-tubulin gene copies in each inoculated wood sample was calculated using the following formula:N = g(d × c)/t × c
where
N = the calculated number of β-tubulin gene copies in 100 mg of wood sampleg = the mean number of gene copies detected by qPCRd = the total gDNA extracted from 100 mg of wood (100 µL)c = DNA concentration (µL)t = the amount of DNA template (5 µL) in one reaction

The amount of pathogen detected in wood sections inoculated with BCA13, BCA14, BCA17, pathogen only (positive control), and SDW (negative control) were compared.

### 3.5. Pseudomonas Colonization and Establishment in Grapevine Tissues

#### 3.5.1. Production of Rifampicin-Resistant Strain

The endophytic colonization of grapevine tissues by *Pseudomonas* was studied using a laboratory-induced rifampicin-resistant (RifMut) strain from BCA17 (*P. poae*). The RifMut strain was developed following the methods of Adorada et al. [72] and West et al. [27]. A single colony of BCA17 was streaked onto NA amended with 1 ppm of rifampicin (Sigma Chemical Co., St. Louis, MO, USA) and resulting colonies subsequently exposed to increasing concentrations of rifampicin (5, 10, 50, and 100 pm). The final *resistant* strain, now referred to as RifMut, was re-inoculated onto NA with 100 ppm rifampicin on five subsequent occasions to ensure the stability of the resistance.

#### 3.5.2. Pre-Planting Application of RifMut Strain

The potential for transmission of strains of *Pseudomonas* in nursery propagation materials was tested by treating grapevine canes with RifMut prior to rooting. To acclimatize them at room temperature, dormant canes (cv. Shiraz) were removed from the 4 °C cold room 24 h prior to RifMut inoculation. All canes were surface sterilized using the methods described earlier.

To prepare the bacterial suspension, a loopful of 2-day-old RifMut culture grown on NA was suspended in 500 mL of ¼ strength Ringer’s solution (Sigma Aldrich, St. Louis, MO, USA) and incubated in an orbital shaker as described previously. The bacterial density was determined after 24 h prior to cane treatment.

For inoculations, canes were trimmed (5 mm length) at the apical and basal ends to create fresh wounds and then placed in a 500 mL beaker containing 250 mL of RifMut suspension with the basal ends of the canes (~4 cm length) immersed in the bacterial suspension. Control canes were placed in a 500 mL beaker containing 250 mL of ¼ Ringer’s solution only. Fifteen replicate canes per treatment were used. All treatments were incubated at 25 °C under a 12:12 dark:light regime. A 500 mL beaker containing 250 mL of ¼ Ringer’s solution without canes was also incubated to account for water loss due to evaporation. After 24 h, the canes were removed from the beakers and the amount of inoculum absorbed by the canes was calculated, after accounting for loss to evaporation.

After inoculation, five canes per treatment were randomly selected and used for re-isolation to determine if the RifMut strain was inside the canes. Tissue samples (~1 cm length) were cut from the base, middle, and top sections of each cane and surface sterilized as described previously. Each section was cut longitudinally and plated onto NA amended with 100 ppm of Rifampicin and observed for bacterial growth following incubation at 25 °C in the dark for 5–7 days. The 10 remaining canes per treatment were rooted in plastic containers containing perlite. After eight weeks, the plants were uprooted, washed, and had their roots removed. The re-isolation of the RifMut was further carried out and observed for bacterial growth as described previously. The recovery of the bacteria was evaluated in the following manner: 0—bacteria not recovered in both cane sections; 1—bacteria were recovered in only 1 section; 2—bacteria were recovered in both cane sections.”

#### 3.5.3. Post-Planting Application of RifMut Strain

The ability of *Pseudomonas* to colonize and establish in potted grapevines was further investigated using RifMut. Dormant cuttings (cv. Shiraz) were rooted and planted in pots in the spring of 2017. For inoculation, fresh wounds were created by drilling through the second internode of the grapevine plant and a 20 µL suspension of RifMut (estimated to be 10^10^ CFU) was pipetted onto the fresh wounds and wrapped with Parafilm^®^M (Merck, Sydney, NSW, Australia). The potted vines with five replicates per sampling period were maintained at ambient temperature (17–27 °C) inside the glasshouse for up to 6 months. The negative control was inoculated with sterile distilled water. The translocation of the RifMut strain within the grapevine tissue was assessed at 1-, 3-, and 6-months post-inoculation.

For each assessment, five replicate vines were uprooted, washed, and had their roots removed. Each vine was cut into 1 cm sections from the apical tip down to the crown and processed as described earlier. At 6 months, further isolations were conducted from the lateral shoots above the inoculation point to determine the lateral movement of the RifMut strain. All wood sections were plated onto NA amended with 100 ppm of rifampicin and observed for bacterial growth as previously described. The recovery of the bacteria was evaluated as described in Section 3.5.2.

### 3.6. Elucidation of Antagonistic Mechanisms of P. poae BCA17 against GTD Pathogens

#### 3.6.1. Bioactivity of *P. poae*-Derived Culture Filtrates on Mycelial Growth and Spore Germination of GTD Pathogens

To prepare the cell-free culture filtrates, *P. poae* BCA17 was streaked on NA then incubated for 24 h, and a loopful from a single colony was transferred into 10 mL of NB with and without the mycelial plug (4 mm) from either *D. seriata* DAR79990, *N. parvum* DAR80004, *N. luteum* DAR81016, or *E. lata* WB052. All cultures were incubated for 24 h in an orbital shaking incubator at 25 °C, 150 rpm. The resulting bacterial suspensions (10 mL) were passed twice through a 0.22 µm Minisart syringe filter (Sartorius Stedim Biotech, Goettingen, Germany) to remove bacterial cells, creating a cultural filtrate. The filtrate was plated onto NA to ensure that it was free from bacterial cells.

##### Effect of Culture Filtrates on Mycelial Growth

To investigate the effects of culture filtrates on mycelial growth, the culture filtrates (10 mL) derived from *P. poae* BCA17 with and without co-incubation with a GTD pathogen were transferred into 55 mm Petri plates. A mycelial plug (4 mm) obtained from the margin of 5-day old cultures *D. seriata* DAR79990, *N. parvum* DAR80004, *N. luteum* DAR81016, and *E. lata* WB052 was added to individual Petri plates containing the culture filtrates and to plates containing only NB, which acted as negative controls. Three replicate plates were used per treatment. All plates were incubated at 25 °C in the dark. After 7 days, the mycelium was harvested and placed on a sterile filter paper (Whatman™, GE Healthcare Life Sciences, Hartlepool, Cleveland, UK). The agar plug was removed, and the mycelia was allowed to air dry for 30 min before being weighed.

##### Effect of Culture Filtrates on Spore Germination

Conidia were produced by transferring mycelial plugs of *D. seriata* DAR79990, *N. parvum* DAR80004, and *N. luteum* DAR81016 onto prune extract agar [73] and incubated following the methods previously described for *N. luteum*. Conidial suspensions for each isolate were prepared, as previously described. An ascospore suspension of *E. lata* isolated from infected grapevine wood was obtained from the South Australian Research and Development Institute, Australia (SARDI). All spore densities were adjusted to 3 × 10^5^ spores/mL. Aliquots (200 µL) of the spore suspensions from each of the pathogens were transferred into a sterile 1.5 mL tube and an equal volume of the BCA17 culture filtrate without co-inoculation with fungal pathogen was added. For the control treatment, an equal volume of NB was added to the spore suspension instead of the culture filtrate. The assay was performed with three replicate tubes per treatment. All tubes were incubated for 24 h at 25 °C. To assess germination, a loopful of the spore suspension was transferred onto a microscope slide. The first 100 spores in the field of view (40×) were counted at 2 counts per isolate and the average number of spores that germinated (with a germ tube half the length of the spore) were recorded.

#### 3.6.2. Identification of Secondary Metabolite Biosynthetic Gene Clusters

The genomes of four strains of *P. poae* (BCA13, BCA14, BCA17, and JMN1) were sequenced, as described by Niem et al. [74]. Strains BCA13, BCA14, and BCA17 were antagonistic towards GTD pathogens, while *P. poae* JMN1 did not exhibit any antagonistic activity [29]. Draft genomes were assembled and annotated as described by Niem et al. [74]. Genome contigs were submitted to the antiSMASH 4.0 web-server [61] to identify gene clusters that may play a role in the control of plant pathogens.

#### 3.6.3. Extraction and Identification of Bioactive Lipopeptides

The detection of antifungal lipopeptides was performed using Matrix-Assisted Laser Desorption/Ionization Time-Of-Flight mass spectrometry (MALDI-TOF MS) analysis. A crude extraction of lipopeptide was achieved following the methods of Sajitha et al. [75]. Two strains, *P. poae* BCA17 (antagonistic) and *P. poae* JMN1 (non-antagonistic), were analyzed for the presence of bioactive lipopeptides. A bacterial suspension (10 µL) of each strain was streaked on one side of a PDA plate, while a mycelial plug of the GTD pathogen *D. seriata* was placed on the opposite side. After 72 h of incubation at 25 °C, one loopful of the bacterial cells from the interface of the dual culture was suspended in 500 µL of acetonitrile with trifluoroacetic acid (0.1%) for 2 min. The mixture was vortexed to obtain a homogenized suspension. Bacterial cells were separated from the supernatant by centrifugation at 5000 rpm for 10 min. The cell-free supernatants were transferred to new microcentrifuge tubes and were sent to the Australian Proteome Analysis Facility (APAF) for MALDI-TOF-MS analysis.

The samples were diluted 1:10 with 50% acetonitrile 0.1% formic acid and directly infused into a QExactive Plus mass spectrometer. MS1 scan indicated the detection of intact lipopeptides, and MS/MS and subsequent de novo sequencing was performed on the ion at 1049.6654 *m*/*z* to assess its sequence.

### 3.7. Data Analysis

All data were analyzed using GenStat (18th edition) (VSN International Ltd., Hemel Hempstead, Hertfordshire, UK) and tested for homogeneity using Levene’s test at *p* ≤ 0.05. The effects of treatments for all the above-mentioned experiments were individually analyzed by one-way ANOVA. Pairwise comparison of treatment means was performed using least significant difference (LSD) at *p* ≤ 0.05. The standard error of the mean was also obtained to estimate the standard deviation of the sampling distribution of the mean.

For the culture filtrate assays, the pathogen growth rates differed, thus the effects of BCA17 culture filtrate on mycelial biomass for each pathogen were determined separately using one-way ANOVA, with significance at *p* ≤ 0.05. Pairwise comparison of means was performed using LSD (*p* ≤ 0.05 significance level). Differences on the effect of BCA17 culture filtrate on spore germination for each species were determined using *t*-tests at *p* ≤ 0.05 level.

## 4. Discussion

The main aim of this study was to investigate the antagonistic ability of strains belonging to *Pseudomonas poae* and *P. moraviensis* against the GTD pathogen *Neofusicoccum luteum in planta* and some of the direct mechanisms of action employed by *P. poae* in the control of GTDs. This study also investigated the colonization capacity and persistence of these bacterial strains when artificially inoculated in living hosts. The results demonstrated that particular strains of *P. poae* could prevent/suppress *N. luteum* infection *in planta*, indicating that they have the potential to be used as BCAs against Botryosphaeria dieback (BD). Further, a diffusible compound, possibly a cyclic lipopeptide, has been identified to exert an antagonistic activity against GTD pathogens in vitro. To our knowledge, this study is the first attempt to investigate *Pseudomonas* spp. endophytic to grapevine for their potential to control the GTD pathogen *N. luteum*.

*In planta* experiments affirmed the efficacy of an antagonistic *P. poae* strain as a BCA. The recovery of the pathogen *N. luteum* was nil to low in the presence of BCA17 in both detached cane and potted vine inoculations. This was further confirmed by a challenge inoculation on potted vines, which resulted in an 80% reduction in pathogen recovery when canes were treated with BCA17. The qPCR analysis further corroborated these results with a significant decrease in the detectable amount of pathogen DNA (40-fold reduction) when the vines were treated with BCA17. A study in New Zealand reported endophytic *Pseudomonas* obtained from stem tissues of *Leptospermum scoparium* (mānuka) demonstrated strong inhibitory activity against *N. luteum* in a dual culture assay [76]. This strain of *Pseudomonas* also reduced the lesion lengths and colonization of the host by *N. luteum* by up to 40% [77]. *P. poae* strain BCA17 from the inner grapevine tissues from this study was shown to have similar antagonistic characteristics to the mānuka strain reported by Wicaksono et al. [77]. The challenge inoculations showed that BCA17 was able to suppress *N. luteum in planta* using detached canes and potted vines and had a demonstrable efficacy as a wound protectant on potted vines. However, this needs to be placed in the context of being applied towards a single GTD pathogen, namely *N. luteum.* Additional experiments are now required to test this BCA efficiency in vineyards and determine its stability and persistence under field conditions and then its efficacy against a wider range of GTD pathogens.

*Pseudomonas poae* has been isolated in a few other hosts, with some strains also exhibiting antagonistic activity against certain plant pathogens. *P. poae* strain DSM 14936 was first reported and described by Behrendt et al. [78] and was isolated from the phyllosphere of grasses (*Poa* spp.). Another strain isolated from the internal root tissue of sugar beet, *P. poae* RE*1-1-14, was found to inhibit the sugar beet root pathogens *Phoma betae*, *Rhizoctonia solani*, and *Sclerotium rolfsii* [79,80]. Field trials using this strain of antagonistic bacteria demonstrated control of late root rot caused by *R. solani* [80].

In this study, *P. poae* strain BCA17 was determined to be an efficient colonizer, capable of multiplying and establishing itself within the grapevine. The RifMut, a derivative resistant to the rifampicin generated from BCA17, was able to endophytically colonize grapevine tissues regardless of whether it was inoculated basally as a pre-planting treatment on cuttings or apically applied on established glasshouse potted vines. The upward movement of bacteria in grapevines has previously been documented for the pathogenic *Agrobacterium tumefaciens*, which has been reported to move up to 30 cm within 24 h when freshly cut basal ends of the shoots are dipped in a bacterial suspension [81]. In grapevine, rapid bacterial colonization via the primary xylem is possible due to the presence of numerous long vessels (up to 1 m), which facilitate ease in bacterial movement within the stem [82]. Thorne et al. [82] further postulated that there are open, continuous vessels from the stem to the leaf lamina of grapevine. As BCA17 has shown promise as a BCA for *N. luteum* in grapevine, there may be potential for this to be used in nurseries where the grapevine cuttings can be pre-treated with the bacterial strain. These findings may be important, especially with the emerging issue of propagation materials being a source of inoculum of GTDs in new vineyards [83,84].

The inoculation of three-month-old potted vines also resulted in the successful colonization and persistence of the bacterial BCA17. Upward and downward movement of the RifMut strain was noted at 1 month post-inoculation and complete colonization 6 months post-inoculation, with the RifMut strain being recovered from the tip of the living tissue down to the basal end of the trunk, below the soil line, and in the lateral shoots. In 1-yr-old and 3-yr-old potted vines, West et al. [27] demonstrated movement of the bacteria *Bacillus cereus* from inoculated leaves down to the vine shoots. However, after 4 weeks *B. cereus* was found only at a 6.5 cm distance from the inoculation point. Wicaksono et al. [77] found a strain of *Pseudomonas* from mānuka was able to colonize the wound site of a 1-yr-old Sauvignon Blanc vine for 6 months. However, the distance travelled by this endophytic *Pseudomonas* within the vine was not reported [77]. In this study, BCA17 was able to persist within the grapevine for longer than 4 weeks and its ability to colonize the grapevine tissue 6 months after introduction implies that it may be a promising BCA. Host colonization by the BCA can be considered another asset for the suppression of plant diseases. To assess the long-term establishment and persistence of BCA17 in the grapevine, further studies are required using mature vines growing in vineyard conditions.

This study also attempted to identify potential compounds that can explain the antagonistic activity of BCA17. The production of bioactive diffusible compounds by BCA17 was evident by the reduction in biomass produced by *Diplodia seriata*, *N. luteum*, *N. parvum*, and *Eutypa lata* when incubated in the cell-free culture filtrate of BCA17. The culture filtrate had a similar inhibitory effect on the spore germination of the above-mentioned GTD pathogens, with a 10–99% reduction in spore germination.

Several antagonistic *Pseudomonas* with a biocontrol ability against a wide range of fungal pathogens have been reported to produce various antifungal compounds [13,20,21,32,33,52,53,85,86,87], including hydrolytic enzymes, such as proteases, cellulases, chitinase, and ß-glucosidase; antibiotics, such as DAPG, pyrrolnitrin, pyoluteorin, phenazines, and siderophores; and several CLPs [88,89]. Cell-free extracts of *Pseudomonas poae* RE*1-1-14 and purified poaeamide were reported to cause the lysis and immobilization of *Phytophthora capsici* and *P. infestans* zoospores but a cell-free culture supernatant of the poaeamide-deficient mutants did not [73].

The draft genomes of *P. poae* strains BCA13, BCA14, BCA17, and JMN1 revealed that all strains possess the same set of genes [67] encoding for pyoverdines (also known as pseudobactins) and for APEs (aryl polyenes), APE Vf, and APE Ec, which comprised the largest biosynthetic gene cluster in the four strains examined here. Pyoverdines are important for the acquisition of Fe^3+^ ions (competition for iron) from the environment and may serve as intracellular signaling compounds controlling gene expression [90]. In general, APEs play a role in protecting bacterial cells from exogenous oxidative stress by reducing the concentration of free radicals that would otherwise cause damage to other cellular lipids, proteins, or nucleic acids [91]. Focusing on the secondary metabolite (SM)-encoding genes present in the genome of strains BCA13, BCA14, BCA17, and JMN1 revealed the presence of biosynthetic genes (BGs) that could be involved in the production of antifungal compounds. Biosynthetic gene clusters (BGCs) for the synthesis of different CLPs were abundant in the genomes of the four strains. CLPs are produced by non-ribosomal peptide synthetases (NRPSs) of bacteria, mostly *Pseudomonas*, *Bacillus*, and *Streptomyces* spp. [88], and are described to perturb membrane function, leading to broad antibacterial and antifungal activities [72]. Among the many CLP-SMs produced by *P. poae* strains BCA13, BCA14, BCA17, and JMN are viscosin [92,93,94,95], orfamide [96], sessilin [96,97], bananamide [98], white-line-inducing principle (WLIP) [99], anikasin [100], and poaemide [73], which have been suggested to play a role in the biological control of plant pathogens [92]. Poaemide produced by *Pseudomonas poae* RE*1-1-14 has been reported to be the determinant factor in the biocontrol ability of the bacteria against sugar beet pathogens [73]. Like other CLPs, this lipopeptide was also essential for the surface motility and swarming ability of *P. poae* RE*1-1-14 and enhanced the root colonization of sugar beet seedlings [73].

While all the SM-BGC coding for the reported antifungal CLP biosynthetic paths are present in both the antagonistic and non-antagonistic *P. poae* strains, it remains unknown if these genes are expressed similarly by the four strains. As part of the continuing effort to understand the biocontrol ability of BCA17, an attempt to identify the CLP profile of the antagonistic and non-antagonistic *Pseudomonas* strains was undertaken using MALDI-TOF analysis. An unknown cyclic lipopeptide only produced by the antagonistic strains of *P. poae* peaked at 2098.2913 *m*/*z* with the molecular formula C_48_H_87_N_11_O_12_ and the amino acid sequence Leu-Val-Gln-Leu-Val-Val-Gln-Leu-Val (LVQLVVQLV). The CLP poaemide from *P. poae* RE*1-1-14 appears to be distinct from the lipopeptide produced by the four strains used in this study, peaking at *m*/*z* 1275.7805 in the high-resolution electrospray ionization mass spectrometry (ESI-MS), with a molecular formula C_61_H_108_N_10_O_17_ and the peptide sequence Xle-Glu-Thr-Xle-Xle-Ser-Xle-Xle-Ser-Xle (Xle = Leu or Ile) [73].

To date, the role of the potentially novel CLP in the biocontrol ability of BCA17 requires further confirmation by in vitro testing with purified compounds and *in planta* assays to assess the effect on pathogen infection and disease development. This can only be elucidated after the careful identification and purification of the compounds. However, this may prove to be challenging as a BCA can employ more than one mechanism to suppress a pathogen. Furthermore, the relative importance of a particular mechanism may be affected by many other factors, including nutrient availability, stress factors, or signal molecules of a microbial or plant origin, which may affect the expression of biocontrol traits. Gene knockout studies targeting the biosynthetic genes may be a more effective way to confirm the importance of these compounds in the biocontrol ability of *P. poae* BCA17. The full potential of any BCA can only be achieved if there is a better understanding of the mechanisms that govern their biocontrol ability. Such knowledge can be useful in providing optimum conditions that will favor production of these antagonistic compounds.

## Figures and Tables

**Figure 1 plants-12-02132-f001:**
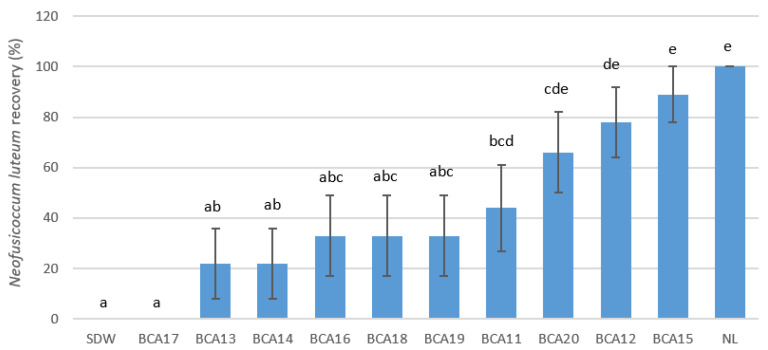
Recovery of the pathogen *Neofusicoccum luteum* (NL) at 4 weeks post-inoculation, from detached canes treated with different strains of *Pseudomonas*. Bars with different letters indicate significant differences in pathogen recovery with the strains of *Pseudomonas* at *p* ≤ 0.05 LSD. Error bars represent standard error of the means. Sterile distilled water (SDW) served as the negative control. BCA strains used for challenge inoculation are BCAs 11, 12, 13, 14, 15, 16, 17, 18, 19, and 20.

**Figure 2 plants-12-02132-f002:**
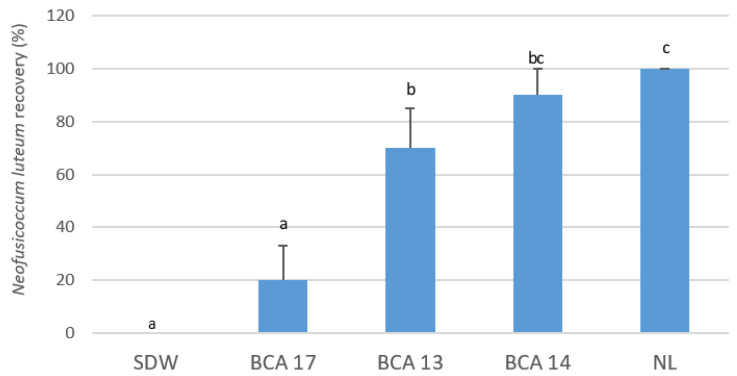
Recovery of the pathogen *Neofusicoccum luteum* (NL) at 3 months post-inoculation, from glasshouse potted vines treated with *Pseudomonas poae* strains BCA13, BCA14, and BCA17 prior to pathogen inoculation. Bars with different letters indicate significant differences in pathogen recovery upon challenge inoculation with the selected BCA strains at *p* ≤ 0.05 LSD. Error bars represent standard error of the means. Sterile distilled water (SDW) served as the negative control. BCA strains used for challenge inoculation in potted vines are BCAs 13, 14, and 17.

**Figure 3 plants-12-02132-f003:**
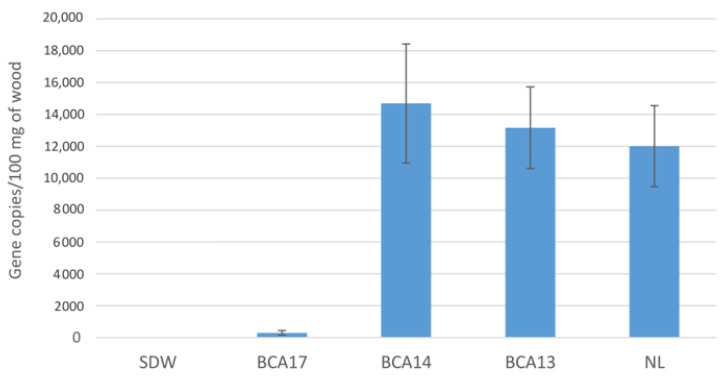
Number of Botryosphaeriaceae-associated β-tubulin gene copies detected by quantitative PCR analysis of DNA extracted from potted vines treated with BCA strains prior to inoculation with *Neofusicoccum luteum* (NL). Each treatment is represented by 100 mg of dried wood from 10 replicate vines, with 4 technical replicates per DNA sample. Bars with different letters indicate significant differences in the number of Botryosphaeriaceae β-tubulin gene copies detected by qPCR at *p* ≤ 0.05 LSD. Error bars represent standard error of the means. Sterile distilled water (SDW) served as the negative control.

**Figure 4 plants-12-02132-f004:**
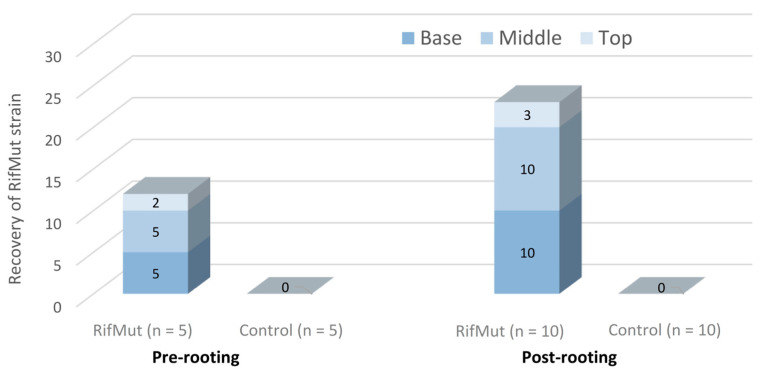
Recovery of the rifampicin-resistant (RifMut) strain of *Pseudomonas poae* BCA17 from dormant canes dipped in bacterial suspensions for 24 h and incubated 1 week (pre-rooting) or rooted for 8 weeks (post-rooting).

**Figure 5 plants-12-02132-f005:**
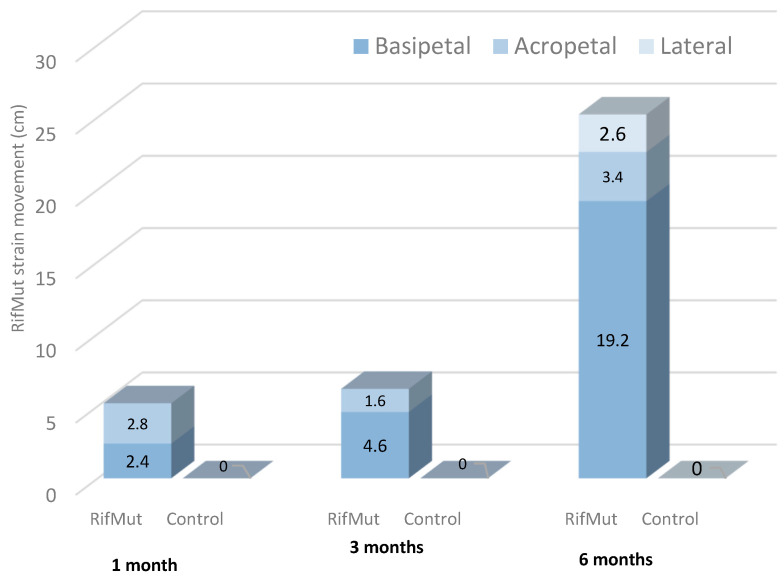
Re-isolation of rifampicin-resistant mutant (RifMut) strain of *Pseudomonas poae* BCA17 from inoculated potted vines at three different sampling periods.

**Figure 6 plants-12-02132-f006:**
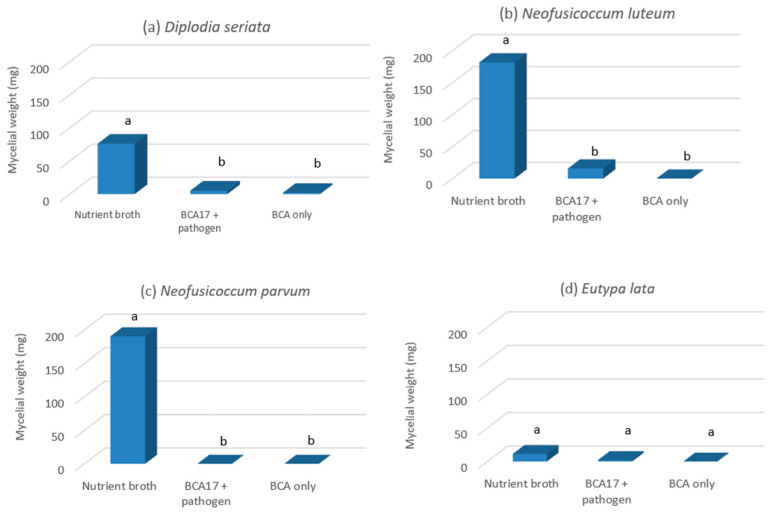
Effect of culture filtrates derived from *Pseudomonas poae* BCA17 on mycelial growth of (**a**) *Diplodia seriata*; (**b**) *Neofusicoccum luteum*; (**c**) *N. parvum*; and (**d**) *Eutypa lata* after 7 days in culture filtrate. The treatments were filtrates from the BCA17 strain nutrient broth culture with (BCA17 + pathogen) and without the pathogen (BCA only). Nutrient broth served as the control. Bars with different letters indicate significant differences in mycelial biomass at *p* ≤ 0.05 LSD.

**Figure 7 plants-12-02132-f007:**
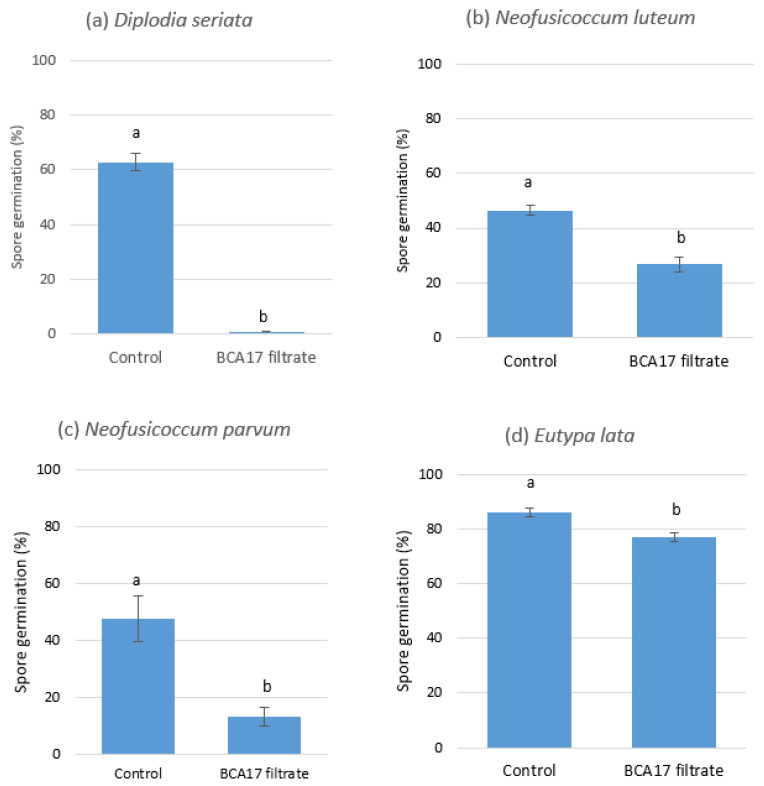
Effect of culture filtrates derived from *Pseudomonas poae* BCA17 on spore germination of (**a**) *Diplodia seriata*; (**b**) *Neofusicoccum luteum*; (**c**) *N. parvum*; and (**d**) *Eutypa lata* after 24 h incubation of the spores in BCA17 filtrate. Nutrient broth served as the control. Bars with different letters for each species indicate significant differences in spore germination at *p ≤* 0.002 using *t*-tests.

**Figure 8 plants-12-02132-f008:**
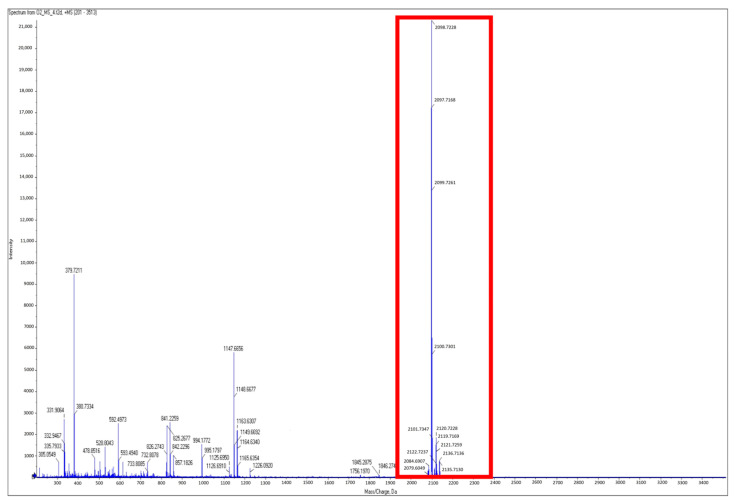
MALDI-TOF spectrum for the antagonistic secondary metabolite produced by *Pseudomonas poae* BCA17. Red box indicates the presence of a dominant lipopeptide at 2098.2913 *m*/*z* ([M + H]^+^ ion).

**Figure 9 plants-12-02132-f009:**
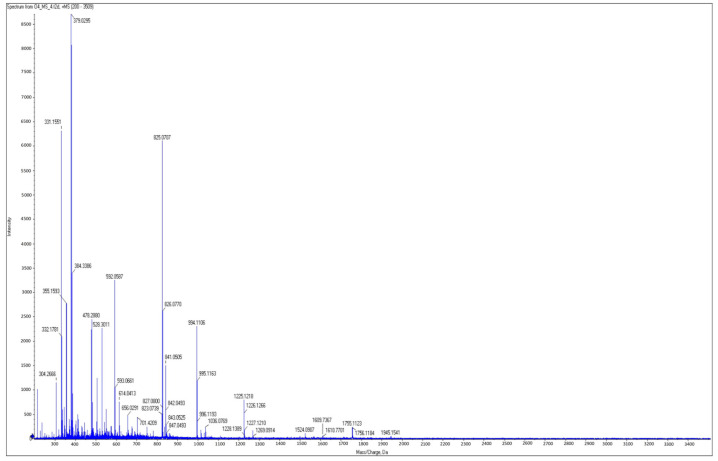
MALDI-TOF spectrum for the non-antagonistic secondary metabolite produced by *Pseudomonas poae* JMN1 indicating the absence of the lipopeptide at 2096.297 *m*/*z* ([M + H]^+^ ion).

**Figure 10 plants-12-02132-f010:**
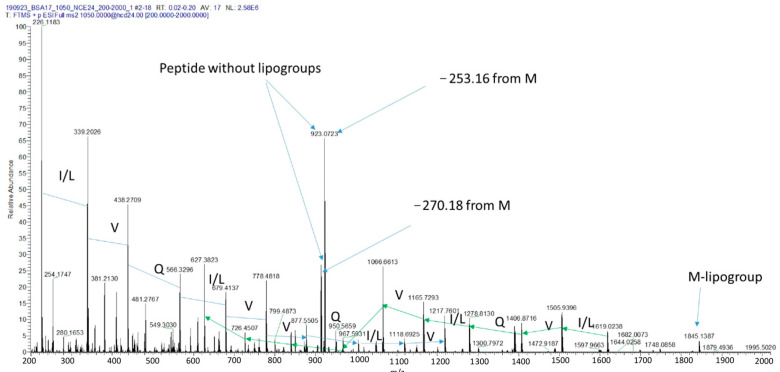
Fragmentation pattern (MS/MS) of the new lipopeptide identified in *P. poae* BCA17 sample. Amino acid sequence information and key ions are labelled on the spectrum.

## Data Availability

The data presented in this study are contained within the article and Appendix A.

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
