# Peer review of "Biocontrol Potential of an Endophytic *Pseudomonas poae* Strain against the Grapevine Trunk Disease Pathogen *Neofusicoccum luteum* and Its Mechanism of Action"

_plants, 2023, doi:10.3390/plants12112132_

Round 1
Reviewer 1 Report
Peer review report on: “Biocontrol potential of endophytic Pseudomonas against selected grapevine trunk disease pathogens, and its mechanism of action”
Recommendation: Major Revision to be fluid and more precise, avoid redundancy, and align with recent bibliography.
Comments:
Grapevine trunk diseases (GTDs), including Botryosphaeria dieback (BD), are a major concern worldwide, due to the heavy economic and agronomic losses they generate. Nowadays, it is still difficult to develop sustainable strategies to control them. Because of the limited number of management options available to control GTDs, biocontrol agents have been intensively explored as a viable alternative.
The authors’ article shows that a Pseudomonas poae isolate (BCA17) effectively suppresses the BD pathogen Neofusicoccum luteum in detached canes and potted vines of cv Shiraz (upon co-inoculation). They also show the ability of BCA17 to colonize and persist in grapevine tissues (useful asset for easy transfer in vineyards, but also for easy propagation in nurseries). Finally, the authors identified a novel secreted compound (a cyclic lipopeptide) that they suspect to be a key determinant of the antagonistic activity of BCA17 against N. luteum.
So these authors provide new knowledge on the direct interaction between BCA17 and Neofusicoccum luteum, the transfer of which could contribute to a better sustainable management of BD in planta.
Using two plant models (detached canes and potted vines), the authors assessed: (1) the beneficial impact of BCA17 on N. luteum in-planta (100% and 80% of protection in co-inoculated detached canes and potted vines, respectively), both by examining fungal growth recovery from co-treated plants and by quantifying the pathogen DNA in-planta. (2) the colonization and establishment abilities of BCA17 in-planta using a rifampicin-resistant strain (re-isolation several days after treatment at different organization levels of the plant). The antagonistic origin of BCA17 against N. luteum was then investigated, by examining: (1) the activity of the BCA17-derived culture filtrate on mycelial growth and conidial germination, (2) the presence of the coding SM-BGCs in the genome of several Pseudomonas isolates, (3) the bioactive lipopeptides differentially produced by antagonistic and non-antagonistic BCA species.
The experimental design is appropriate, and the related methods can be further clarified using the comments in the attached pdf file.
It should also be noted that: (A) There are lacks in the bibliography to adequately introduce this topic (especially recent references, despite the 104 references cited), and then to develop a serene discussion. (B) Be careful to not generalise the information related to one BD pathogen to the multiple GTD pathogens.
à Therefore, the current study provides a credible- hypothesis and data regarding the efficacy of BCA17 in a direct control of the BD pathogen N. luteum on detached canes and potted vines of cv Shiraz (after co-inoculation), possibly via a novel cyclic lipopeptide
THUS: Despite the redundancies and missing references, the article is quite well written and described. The authors seem to have carried out the experiments and their analysis with care. However, I suggest several changes as indicated directly in the attached pdf. I also ask for some clarifications, adds and ask different questions directly in the attached pdf. Thanks to the authors for their interesting work and for their future answers.

as indicated above and in the attached pdf file
Reviewer 2 Report
The manuscript by Niem and colleagues describe the characterization of a P. poae strain as putative biocontrol agent to fight grapevine trunk diseases (GTD), particularly against the pathogen N. luteum. A putative role of a novel polypeptide as antifungal agent is discussed
Although the work is technically well executed and of potential interest to the sector, it suffers from a number of shortcomings in its presentation (listed below) that should be corrected.
Major comments:
Ø The title of the manuscript should be changed as it clearly misleads us about what we will find in the manuscript. I think that a tentative Title like “Biocontrol potential of an endophytic Pseudomonas poae strains against the grapevine trunk pathogen Neofusicoccum luteum and its mechanism of action” could be considered by the authors.
Ø Abstract should be widely reorganized and shortened to be more concise. In lines 22-24 the authors indicate “Endophytic strains of Pseudomonas were identified from grape-22 vines in Australia with antagonistic activity toward pathogens associated with Botryosphaeria die-23 back (BD), Eutypa dieback (ED), and Petri disease (PD) in vitro.”. However, these are results from a previous manuscript (reference 30) and accordingly this paragraph should be removed.
Ø Abstract: lines 26-27. The authors state: “(2) the ability of the strains to colonize and persist within grapevine tissues; and (3) 26 the mode of action of the strains in antagonism of N. luteum.”. This statement is incorrect since only the strain BCA17 (P. poae) has been used in these kind of studies, whereas from reading this paragraph it appears that such studies have been carried out on several strains, which is incorrect.
Ø Introduction: lines 57-62. I think this paragraph should be removed since this manuscript is not dealing at all with E. lata, but to the pathogen N. luteum. I think the authors should focus on introducing previous studies on control of GTDs with endophytic bacteria, such as those conducted by Álvarez-Pérez et al., 2017; Appl. Environ. Microbiol. 83 (24): e01564-17 and Martínez-Diz et al., 2021, Pest Manag Sci. https://doi.org/10.1002/ps.6064, among others, instead to review biocontrol of GTDs with fungi like Trichoderma.
Ø Introduction: lines 104-109. Again, the authors should be more precise as objectives 2 and 3 have only been evaluated for the BCA17 strain.
Ø Lines 401-403. Please, indicate percentage of recovery of N. luteum from canes treated with BCA17 since it appears from the observation in Figure 1 that the percentage is 0% (is that right?).
Round 2
Reviewer 1 Report
Dear colleagues,
Thanks to the authors for their interesting work and for their answers.
All the required clarifications have been made in this revised version.
I therefore propose to ACCEPT the manuscript, even if I still suggest to the authors additional minor changes in order to make the manuscript more fluid in some places.
Sincerely

quality of english is ok